# Organophilized Montmorillonites as Fillers for Silicone Pressure-Sensitive Adhesives

**DOI:** 10.3390/ma16030950

**Published:** 2023-01-19

**Authors:** Adrian Krzysztof Antosik, Karolina Mozelewska, Magdalena Zdanowicz, Konrad Gziut, Piotr Miądlicki

**Affiliations:** 1Department of Chemical Organic Technology and Polymeric Materials, Faculty of Chemical Technology and Engineering, West Pomeranian University of Technology in Szczecin, Piastow Ave. 42, 71-065 Szczecin, Poland; 2Center of Bioimmobilisation and Innovative Packaging Materials, Faculty of Food Sciences and Fisheries, West Pomeranian University of Technology in Szczecin, Janickiego 35, 71-270 Szczecin, Poland; 3Department of Engineering of Catalytic and Sorbent Materials, Faculty of Chemical Technology and Engineering, West Pomeranian University of Technology in Szczecin, Piastow Ave. 42, 71-065 Szczecin, Poland

**Keywords:** adhesives, adhesion, cohesion, organophilized montmorillonite, silicone pressure-sensitive adhesives

## Abstract

In the presented work, organophilized montmorillonites (OMMT) with selected quaternary ammonium compounds (QAC) with different chemical structure ((trioctylmethylammonium chloride—A336, dimethyloctadecyl[3-trimethoxysilyl)propyl]ammonium chloride—D, cetyltrimethylammonium bromide—CTAB, 2-methacryloxyethyltrimethylammonium chloride—MOA) were obtained and used as fillers for physically modified silicone pressure-sensitive adhesives (Si-PSA). Before OMMT addition into Si-PSA matrix, they were analyzed via TGA and XRD techniques. Type of chemical structure of QAC affected d-spacing of OMMT. New self-adhesive materials were obtained based on prepared Si-PSA compositions by adding the obtained fillers to the polymer matrix. New tapes exhibit a good level of useful properties as adhesion, cohesion, and tack—the values did not change or slightly decreased; in addition, the tapes with addition of OMMT showed high thermal resistance reaching the measuring limit of the test equipment—to 225 °C.

## 1. Introduction

An adhesive is a material applied to bind surfaces permanently or temporarily. The process is based on a balance of two forces: the cohesive force and the adhesive force [1]. The method of bonding with the usage of adhesive tape is shown graphically in Figure 1. The bonding layer consists of an adhesion and cohesion zone. The interfacial forces between the substrate and the adhesive are the adhesive forces, while the cohesion force are the forces inner adhesive layer [2].

The binding mechanism can be divided into stages following one another. First, the pressure-sensitive adhesive (PSA) on the tape makes contact with the substrate with minimal pressure. Then, the adhesive increases its surface area and passes (penetrates) to the surface of the substrate. In the final stage, the adhesive forms a strong bond, bonding to the substrate [3,4].

In the case of adhesives working on the principle of adhesion, the adhesive fluid turns into a solid after sticking (i.e., making a connection). In the case of PSA, the adhesive remains fluid even after the bond is established. As a result, its resistance to detachment is moderate, and the joint may delaminate, in most cases, without destroying the elements of the stuck materials [5].

Pressure-sensitive adhesives are used in various industries: in packaging, healthcare, electrical insulation, labels, and sign tapes [6]. However, they exhibit relatively poor mechanical and thermal properties, which greatly reduce their use, especially for specialized applications [7,8]. There are many scientists in the world who modify PSAs to improve their properties [9,10,11]. For example, Kajtana et al. prepared nanocomposite acrylic PSA cross-linking under the influence of UV radiation and tested various amounts of modified and unmodified montmorillonite. As a result, they obtained adhesives with increased cohesion [12]. Pang and his team described the improvement of the thermal resistance of adhesives cross-linked with ultraviolet radiation and modified with silica nanoparticles [13]. Pokerznik et al. described lactic acid with glucosyl groups as a monomer that improves the mechanical properties of PSA [14]. Park and co-workers obtained a silicone-acrylate adhesive using a silicone macro-azo-initiator and modification with a silicon urethane dimethacrylate oligomer. As a result, they obtained an adhesive with high thermal resistance and facilitated the wettability of rough surfaces [15]. In order to meet the growing demand for pressure-sensitive adhesives, there is still work to be done to improve the thermal and mechanical properties.

Silicone pressure-sensitive adhesives, due to their specific construction—practically no functional groups—are considered materials for special applications. Their uniqueness makes them often better materials for use in conditions difficult for self-adhesive tapes (e.g., increased humidity, high insolation) than their counterparts made of carbon polymers, e.g., acrylate pressure-sensitive adhesives. The lack of reactivity also makes it difficult to modify these adhesives. In the most common modifications, radical cross-linking (chemical modification) or the addition of fillers are used to obtain specific, facilitated properties (physical modification). From our own experience and literature, it is known to introduce silica-derived fillers such as silica, montmorillonite or kaolin into the polymer matrix, which usually improves the cohesive properties and thermal stability, which is associated with thermal resistance of the composite [16,17,18,19]. New activities in this group of adhesives are the modification of introduced fillers, thanks to which better compatibility between filler and polymer matrix are obtained.

Montmorillonite (MMT) is commonly used as a filler for polymer (nano)composites due to its availability, low costs, and some particular properties. MMT is layered silicate consists of aluminum octahedron sheet sandwiched between two silicon tetrahedron sheets [20]. In the production of adhesives, the nanoclay MMTs were used as filler/reinforcement agent [21,22]. The weak van der Waals and electrostatic forces hold the aluminosilicate layers together and the interlayer distance is dependent on the cation presence and its degree of hydration [23]. Due to the hydrophilic surface of MMT, it is incompatible with many hydrophobic polymers and one of the methods to overcome this problem is modification by exchanging the inorganic cation with organic ions leading to organophilized MMT (OMMT). Modifiers, e.g., with ammonium cation are able to exchange sodium cation and often due to their greater size of the molecule they can lead to an intercalation increasing the clay interlayer. The intercalation mechanisms can be explained through electrostatic interactions, secondary bonding, or covalent bonding [24]. Dispersion degree of intercalated MMT can be improved in more hydrophobic polymer matrix due to, e.g., a presence of the more hydrophobic organophilizer between clay platelets and enlarged clay d-spacing.

The aim of the work was to organophilize sodium montmorillonite with selected ammonium compounds (QAC) and study the influence of OMMT on silicone pressure-sensitive adhesives (Si-PSA). Organophilization efficiency was analyzed with XRD, FTIR, and TGA as well as viscosity of silicon resin/OMMT in time was measured (potlife test). An influence of OMMT addition in Si-PSA systems was investigated carrying out adhesion, cohesion, tack, and shrinkage tests. Moreover, thermal stability as shear adhesion failure test (SAFT) of composite materials was determined.

## 2. Materials and Methods

### 2.1. Materials

Sodium montmorillonite (MMT-Na)-Cloisite Na+ with cation exchange capacity 92 mmol/100 g was purchased from BYK-Chemie GmbH (Wesel, Germany), Quaternary ammonium compounds (QAC): Aliquat^TM^336 (trioctylmethylammonium chloride)—A336—Sigma-Aldrich (St. Louis, MO, USA), dimethyloctadecyl[3-trimethoxysilyl)propyl]ammonium chloride (60% in methanol)—D Acros Organics, cetyltrimethylammonium bromide (>99%)—CTAB Roth; 2-methacryloxyethyltrimethylammonium chloride (70% in water)—MOA Polysciences Inc. (Warrington, PA, USA). Chemical structure of QAC used for MMT modification is presented in Figure 1.

Silicone resin type MQ—SilGrip PSA595 was purchased form Momentive, Waterford, NY, USA, solvent—toluene (Carl Roth, Karlsruh, Germany), cross-linking agent—dichlorobenzyl peroxide—DClBPO (Gelest, Morrisville, PA, USA).

### 2.2. Preparation of Adhesives Composition

In the first stage of research, the starting pressure-sensitive adhesive modified with a thermal cross-linking compound dibenzoyl peroxide in the amount of 1.5 pph (the result of the authors’ previous research [17,18] and with toluene to 50% solids was characterized. The resulting composition was mixed by hand and then different fillers were added from 0.1 to 3 pph per 100 pph of resin.

### 2.3. Preparation of Self-Adhesive Tapes

To obtain a one-sided self-adhesive tape necessary for the tests, the pressure-sensitive adhesive was coated on a polyester foil (foil weight: 50 g/m^2^). Using a semi-automatic coater, a composition having a basis weight of 45 g/m^2^ was coated and put into an oven to cross-link and evaporate the solvent.

### 2.4. Montmorillonite Modification

Medium for modification was selected based on character of organophilizing QAC. Methanol was used for more hydrophobic, water-insoluble agents, i.e., A336 and D. The methodology of MMT modification is presented in Figure 2.

Organophilization in aqueous medium: MMT-Na was dispersed (wt%) in distilled water, in the next day dispersion was diluted to 5 wt% of the clay and mixed for 30 min at 60 °C with 400 rpm and then liquid quaternary ammonium agent (MOA, CTAB) was drop-wise added to the slurry. A rapid increase of the system was observed, thus stirring speed was increased above 1000 rpm for few seconds. Then, the mixture was stirred for 5 h, filtrated (each time the clay was immersed and mixing with water for 15 min) with distilled water and dried in the drier (Binder) at 60 °C until wholly dried product was obtained.

Ogranophilization in methanol: MMT-Na was dispersed (10 wt%) in methanol, for 1 h at ambient temperature with 400 rpm and then liquid quaternary ammonium agent (A336, D) was drop-wise added to the slurry. Then, the mixture was stirred at 45 °C for 5 h, filtrated (each time the clay was immersed and mixing with methanol for 15 min) and dried in the drier to dried product.

### 2.5. XRD Analysis of the Fillers

XRD analyses were performed in order to determine d-spacing of the modified montmorillonite samples according the method presented in Li et al. Work [25]. The X-ray diffraction (XRD) patterns of the catalysts were recorded by an Empyrean PANalytical (Malvern, UK) X-ray diffractometer using Cu K (λ = 0.154 nm) as the radiation source in the 2θ range 1–15° with a step size of 0.052.

### 2.6. Thermogravimetric Analysis (TGA)

Thermal characterization of organophilized fillers was carried out using TGA (Q500, TA Instruments, New Castle, DE, USA). Tests were performed on platinum pans under 50 mL/min air flow, in the temperature range of 40–900 °C at a heating rate 10 °C/min in air atmosphere.

### 2.7. Fourrier Transform Infrarred Spectroscopy of the fillers

Fourier transform infrared (FT-IR) analysis carried out using a FT-IR spectrophotometer (Perkin Elmer Spectrum 100, Waltham, MA, USA), operated at a resolution of 4 cm^−1^, with twelve scans. The spectra were recorded at a wavelength of 4000–600 cm^−1^.

### 2.8. SEM

Obtained adhesives films with filler was analyzed using a scanning electron microscope (SEM). The microscopic analysis was performed using a Vega 3 LMU microscope (Tescan, Brno-Kohoutovice, Czech Republic). An analysis was performed at room temperature with a tungsten filament and an accelerating voltage of 10 kV was used to capture SEM images. All specimens were viewed from above [26,27].

### 2.9. Potlife Tests

Potlife is a parameter that determines the service life of the adhesive composition—that is, the time in which it can be applied. It is referred to as the time needed to double (for higher viscosity) or fourfold (for low viscosity) increase in the viscosity of the original mixture. The test is carried out at room temperature (23 °C) and the measurement is started immediately after mixing the composition. The test was performed using a DV-II Pro Extra viscometer (Brookfield, New York, NY, USA) [28].

### 2.10. Preparation of One-Side Tapes by Using Silicone Pressure-Sensitive Adhesives

The commercial silicone resin (SilGrip PSA595) was mixed with the fillers (montmorillonite without modification and modified montmorillonites) and a cross-linking compound (1.5 pph 2-4-dichlorobenzoyl peroxide—DClBPO per 100 pph of silicone resin) until a homogeneous consistency was obtained. The influence of the filler addition on the tapes adhesive properties was investigated with an addition of different concentrations of the fillers 0.01, 0.1 and 1 pph per resin—selection of the filler amount on the basis of previous tests. The prepared composition was casted on a polyester film with a thickness of 50 g/m^2^ using a semi-automatic PSAT coater (designed in West Pomeranian University of Technology in Szczecin, Poland). Then, the adhesive film was placed in a drying channel for 10 min at 110 °C to cross-link the adhesive film. Then, the cross-linked adhesive film was pulled out and secured with film tape to obtain an adhesive tape with an adhesive thickness of 45 g/m^2^.

### 2.11. Adhesion Determination

The value of adhesion (peel adhesion) to steel at an angle of 180° was determined according to the AFERA 4001—international standard developed by the European Association des Fabricants Europeens de Rubans Auto-Adhesifs—AFERA) using the Zwick/Roell Z010 testing machine (Zwick/Roell, Ulm, Germany). A one-sided Si-PSA film with dimensions of 175 × 25 mm was applied to the degreased steel plate and pressed with a rubber roller weighing 2 kg in order to improve the wettability of the substrate by the adhesive. The test was performed 20 min from the application of the film to the plate with a peeling speed of 300 mm/min. The test was performed five times for each film [29,30].

### 2.12. Cohesion Determination

The value of cohesion was determined in accordance with the AFERA 4012 international standard using a device designed by the International Laboratory of Adhesives and Self-Adhesive Materials of the West Pomeranian University of Technology in Szczecin, which enables automatic measurement of the time of joint crack occurrence. A one-sided adhesive film was applied to the degreased steel plate to form a 25 × 25 mm (6.25 cm^2^) joint and pressed with 2 kg rubber roller to improve wettability. The 1 kg weight was attached to the free end of the film. The setup was then placed in a tripod so that the force of gravity exerted on the weld at an angle of 180°. The cohesion value was defined as the time needed for the weld to crack. The test was carried out at a temperature of 20 °C and 70 °C [31,32].

### 2.13. Shear Adhesion Failure Test (SAFT)

The shear adhesion failure test (SAFT) was performed to evaluate the heat resistance and shear strength of the Si-PSA films. Failure temperature was measured using a shear tester designed at the International Laboratory of Adhesives and Self-Adhesive Materials of the West Pomeranian University of Technology, Szczecin. A 2.5 × 2.5 cm area of PSAT was adhered to stainless-steel plate with use of the 2 kg roller. Then, the weight of 1 kg was hung on the adhesive tape and this system was placed in the machine. The test was carried out in the temperature conditions of 20–250 °C with a temperature increase of 0.5 °C/min. The temperature at which the adhesive failed was recorded as the shear failure temperature [33].

### 2.14. Tack Determination

The tack value was determined by the loop method in accordance with the AFERA 4015 international standard using the Zwick/Roell Z010 testing machine (Zwick/Roell, Ulm, Germany). The Si-PSA film with dimensions of 175 × 25 mm was mounted in the upper grips so as to obtain loops with the adhesive layer on the outside. The sample was lowered perpendicularly to the degreased steel plate placed in the lower jaws, at a speed of 100 mm/min. The contact area was about 6.25 cm^2^. The software recorded the force needed to detach the adhesive film after a short contact with the steel surface, without external forces. The final result was the average of three consecutive measurements [34,35].

### 2.15. Shrinkage Measuremnts

The shrinkage is a dimensions change (decrease) of the material at an elevated temperature with time. It is measured as a percentage or millimeter change in the dimensions of the PVC or PET foil covered with the tested adhesive. After the adhesive has been cross-linked and the foil has been attached to the aluminum plates, the sample is conditioned for 8 weeks at 70 °C. Shrinkage greater than 0.5% or 0.5 mm exceeds the allowable value [36].

## 3. Results and Discussion

Table 1 shows molecular weight of quaternary ammonium compounds used as organophilized agents for sodium montmorillonite. All selected QAC intercalated the clay layers (Figure 2). However, there is no relation between the size of the quaternary compound molecule and its content between clay layers. The structure and chemical character of the organophilizing agent is more likely to influence on the intercalation and enhancing d-spacing (Figure 1). The highest d-spacing interlayers were obtained for QAC possessing the long single alkyl chains: OMMT-CTAB (d_001_2.03 nm) with hexadecyl alkyl chain (C16) and for OMMT-D (d_001_2.48 nm) with octadecyl alkyl (C18) chain and (triethoxysilyl)propyl group. The longer alkyl chain in the compound’s structure, the higher content of the modifier in OMMT and its layer d-spacing (up to 2.48 nm). Despite the molecular weight of A336 being quite high (404.16 g.mol) amongst other QAC, the d-spacing is only 1.75 nm and their content in OMMT is ca. 19%. This could be caused by a spherical hindrance of the three alkyl chains of the A366 as well as by the organophilization method. The modification was prepared in methanol due to the QAC solubility where the clay swelling degree is lower than in aqueous media.

Figure 3 shows FTIR spectra for organophilizing agent (MOA) pristine and modified clay. For MMT-Na, there is a broad peak at 3412 cm^−1^ assigned to hydroxyl group stretching vibrations of silanol in clay interlayer and peak at 999 cm^−1^ is attributed to Si-O stretching vibrations and there were no changes after modification [23]. For OMMT-MOA sample, there are peaks at 1726, 1477, 1322, and 1298 cm^−1^ that came from the organophilizer. This confirms effective MT modification and the presence of ammonium salt in the clay interlayer. The peaks from modifiers were also observed in the case of other OMMTs.

Figure 4 shows selected SEM photos of modified adhesive films in order to present the adhesive and cohesive interaction of the filler in relation to the polymer matrix. The surface of the adhesive after removing the protective layer showed evenly distributed holes, which is caused by stress and the effect of adhesive stretching when removing this layer, resulting from the good adhesive properties of Si-PSAs to materials with low surface energy [37,38] The SEM pictures in Figure 4d show the phase separation of the adhesive with the filler, where the filler particles are well embedded in the polymer matrix, which confirms the adhesive type of matrix–montmorillonite connection [38,39]. Enlargements b and f show fillers under the layer of pressure-sensitive adhesive. The photos did not show increased agglomeration of the filler, the differences between the particle sizes could be caused by the type of modifications carried out on the fillers.

Figure 5 shows the peel adhesion of the tapes. The adhesion for unmodified Si-PSA is 13.35 N/25 mm. In the case of the lowest concentration of the filler, for MMT-Na and OMMT-MOA, adhesion is slightly higher than pure Si-PSA but it decreased with the increase of the fillers amount and this phenomenon was obtained for all composite materials. This is a relatively popular phenomenon in the technology of adhesives production and self-adhesive materials and it can be explained by the fact that when the filler is added, an increase in surface tension is observed so that the compositions achieve lower wettability of the adhesive on the substrate, which indicates a decrease in peel adhesion [40].

In the case of the lowest concentration applied, the highest value was obtained for pristine sodium montmorillonite, while for the content of 0.1 pph, the higher value was obtained for montmorillonite organophilized with MOA (2-methacryloxyethyltrimethylammonium chloride). The lowest values were obtained for MMT-CTAB and MMT-D. The mentioned fillers were organophilized with the QAC with the longest alkyl chains, that could affect a decrease in the peel adhesion.

Figure 6 shows the tack results for adhesive compositions with the fillers. The tack for pure Si-PSA is 22.3 N. As in the previous case, the value of the tack decreased with increasing filler content in the sample. After adding even a small amount of the filler, the silicone PSA began to cross-link, and its structure became stronger and more compact, causing the decrease in stickiness [11]. The tack values for the sample with the OMMT-CTAB were similar to the Si-PSA with unmodified filler and it exhibited the poorest properties.

On the other hand, other organophilized MMT improved the stickiness properties up to approximately 19 N (for OMMT-A336). This may be due to the higher d-spacing of the clay layers and the more hydrophobic character of the additive that can provide fine dispersion and increase the filler-polymer interaction, resulting in less adhesive stress relaxation [41]. The lower content of the filler in the sample, the higher the value of tack parameter. This could be a result of dispersion degree and forming of agglomerates for higher filler concentrations [42].

The results of the cohesion tests at the temperature of 20 and 70 °C are presented in Table 2. The designation 7358-0 denotes the adhesive without the filler. In the case of the starting adhesive, the cohesion at temperature of 20 °C achieved the maximum result (more than 72 h). None of the modifications affected negatively on this feature value. The cohesion at 70 °C for a filler-free adhesive was relatively low (less than two hours) and virtually every addition of filler increases the cohesion value to its maximum value. Only for the modified sample with OMMT-CTAB, it was 30 h. This may be due to the fact the montmorillonite samples were organophilized in this case with one of the compounds with the longest alkyl chain, which resulted in a lower cohesion as in the previous cases and at lower concentrations [18,43]. The reasons for the strength in elevated temperature increase were named as: cohesion strength increase and changes in the level and distribution of residual stresses after nanofiller incorporation [44,45].

The results of the SAFT test for adhesive films before modification (marked with 0) and after modification with the fillers are depicted in Figure 7. The test was carried out from room temperature to 225 °C. The tape samples made from the starting adhesive reached the temperature of 175 °C. It follows that each filler addition increased the thermal resistance of the tapes to the temperature above 205 °C. For samples containing pristine MMT, the maximum value was achieved with the 0.1 pph content. A similar relationship was observed for the sample with OMMT-MOA. Moreover, the results obtained with adhesive films with the presence of MMT-CTAB show maximum values (225 °C) for the initial concentrations, and the worst for the content of 1 pph. This may be due to high filler concentration, causing a decrease in the intrinsic strength of the adhesive and the sample falls off the tile at lower temperatures. On the other hand, samples with OMMT-A336 (trioctylmethylammonium chloride) reached the highest value for the highest filler content. This may be due to the high molecular weight of the modifier (404.16 g/mol). OMMT-D was characterized by the highest amount of modifier in montmorillonite (28.9%). Moreover, the filler was organophilized with a compound with the longest aliphatic chains. Consequently, the values of the thermal resistance are lower than in the previous cases, yet the results are nevertheless much higher compared to the starting adhesive without the filler [42]. It may also reduce the filler’s tendency to agglomeration, which will be manifested by an increase in the thermal resistance of the entire adhesive film [43,45,46].

Figure 8 shows the results of viscosity-pot life for pressure-sensitive adhesives modified with the montmorillonite. The “0 pph” marking in Figure 8a represents the result for the adhesive without any filler. Virtually every modification increases the viscosity of the adhesive. This may be due to an increase in surface tension in the adhesive sample after an addition of filler. The lowest value was obtained for the adhesive samples with the filler: OMMT-CTAB with a hexadecylalkyl chain. This may be due to the modification of the filler with the compound with long aliphatic chain. The highest values were obtained for pristine MMT-Na, and the results most like these results were obtained for resin with OMMT-D. The reason for such phenomenon may be the chemical structure of the modifier in the montmorillonite sample containing trimethoxysilyl group [16,18]. In addition, it may be due to the fact that the introduced modifying material influences the content of silicon material in measured samples (it significantly reduces it); therefore, compared to samples modified with unmodified material, they showed lower shrinkage [44]. Lower viscosity influenced with OMMT addition can facilitate application of the silicone resin on the polymer tapes.

Figure 9 shows the shrinkage results of Si-PSA. Again, “0 pph” marks the sample without modification. The unmodified Si-PSA exhibited very high shrinkage, over 1.3%. Such a high value is unacceptable in the technology of pressure-sensitive adhesives (the acceptable technological value is 0.5% and is shown in the Figure 6 as a straight line). An introduction of clay fillers into the silicone resin resulted in great reduction of the shrinkage value. In the case of the MMT-Na, all the concentrations led to shrinkage below 0.5%. The highest shrinkage for sample with this filler was obtained with concentration of 0.1 pph, which is a different phenomenon than in comparable amount of the other composite samples. This may be due to the lack of any organophilization of the nanoclay. Moreover, all the concentrations used are clustered close to each other. Similar relationships are observed for OMMT-CTAB and OMMT-D. In their case, none of the values exceed the 0.5% limit. However, in the case of the samples with OMMT-MOA, OMMT-A336, the shrinkage values for 0.01 pph concentrations exceed the allowable shrinkage after approximately 70 h. The higher content of the filler, the lower shrinkage value. The lowest shrinkage values were obtained for Si-PSA with 0.1 and 1 pph of OMMT-A336. This could be influenced by higher content of the organophilizer due to higher content of OMMT and their chemical structure. A336 has 3 long alkyl chains that could be a spherical hindrance of the shrinkage process.

## 4. Conclusions

Organophilized montmorillonites (OMMT) with selected quaternary ammonium compounds with different chemical structure ((trioctylmethylammonium chloride—A336, dimethyloctadecyl[3-trimethoxysilyl)propyl]ammonium chloride—D, cetyltrimethylammonium bromide—CTAB, 2-methacryloxyethyltrimethylammonium chloride—MOA) were obtained, characterized (TGA, XRD) and added as Si-PSA fillers.

As a result of the research, novel, innovative composite materials were obtained as one-sided self-adhesive tapes based on silicone glue with increased thermal resistance. The newly produced materials with OMMT retained good performance properties of the adhesion, cohesion, tack, shrinkage, and viscosity, while increasing temperature resisting from 148 °C to 220–225 °C. The adhesion, cohesion and tack values decreased with the increase of filler concentration in Si-PSA. For economic reasons, a great advantage is a preparation of materials with improved thermal properties, adding small amounts of a filler. Low concentration of the fillers can be dispersed in polymer matrix without their agglomeration.

Simple physical modification of adhesives also increases their use in production lines, i.e., fillers addition decrease viscosity than can facilitate application of silicone resins onto liners as well decrease shrinkage of the Si-PSA.

Materials developed in this way can be successfully used as connecting elements in heating, exposed to high temperatures, as well as in heavy industry, automotive, and aviation, or even as masking tapes in the painting industry.

## Data Availability

Not applicable.

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
