# Peer review of "Organophilized Montmorillonites as Fillers for Silicone Pressure-Sensitive Adhesives"

_materials, 2023, doi:10.3390/ma16030950_

Round 1

Reviewer 1 Report

This report presents the use of organophilized montmorillonites as fillers in silicone pressure-sensitive adhesives. The manuscript is well-structured. I recommend publication after minimal revisions.

1.         The abstract can be improved to reflect montmorillonite modification.

2.         The molecular formulae of the four modifiers might be drawn to help the reader understand the variations in the molecular structure of the modifiers.

3.         Montmorillonite layer spacing is connected not only to the molecular type of modifier, but also to the quantity of modifier. Is the amount of modifier utilized in this study sufficient to accomplish maximal intercalation modification? Is the degree of alteration linked to adhesion?

4.         It would be extremely helpful to clarify the intercalation mechanism if some references to discuss the molecular space conformation of intercalators could be included. Materials Chemistry & Physics, 2012, 135(2-3):681-686.

5.         Is it more favourable to intercalation modification with calcium soil for sodium-soil modification?

6.         The image may have been improved.

Author Response

The authors would like to thank to the Reviewer and truly appreciate his comments, questions and corrections. Please find the detailed answers below.

This report presents the use of organophilized montmorillonites as fillers in silicone pressure-sensitive adhesives. The manuscript is well-structured. I recommend publication after minimal revisions.

  1. The abstract can be improved to reflect montmorillonite modification.

The abstract has been changed as suggested by the reviewer, we hope it is now correct.

  1. The molecular formulae of the four modifiers might be drawn to help the reader understand the variations in the molecular structure of the modifiers.

The molecular formulas of selected quaternary ammonium compounds have been added to Section 2.1.

  1. Montmorillonite layer spacing is connected not only to the molecular type of modifier, but also to the quantity of modifier. Is the amount of modifier utilized in this study sufficient to accomplish maximal intercalation modification? Is the degree of alteration linked to adhesion?

The modifier amount used for MMT organophilization was calculated based on cation exchange capacity of the clay with some excess of the modifier.

The supplementary part has been added to this section.

„The longeralkyl chain in the compound’s structure the higher content of the modifier in OMMT and its layer d-spacing (up to 2.48 nm). Despite the molecular weight of A336 is quite high (404.16 g/mol) amongst other QAC, the d-spacing is only 1.75 nm and their content in OMMT is ca. 19%. It could caused by a spherical hindrance of the three alkyl chains of the A366 as well as by organophilization method. The modification was prepared in methanol due to the QAC solubility where the clay swelling degree is lower than in aqueous media.”

  1. It would be extremely helpful to clarify the intercalation mechanism if some references to discuss the molecular space conformation of intercalators could be included. Materials Chemistry & Physics, 2012, 135(2-3):681-686.

The part with expanation and additional reference has been added: „Modifiers with e.g. ammoniumcationareable to exchange sodiumcation and often due to their greater size of the molecule they can lead to an intercalation increasing the clay interlayer. The intercalation mechanisms can be explained through electrostatic interactions, secondary bonding or covalent bonding [Takashi]. A dispersiondegree of intercalated MMT can be improved in the more hydrophobic polymer matrix due to increased clay d-spacing and presence of e.g. of the more hydrophobic organophilizer.

  1. Is it more favourable to intercalation modification with calcium soil for sodium-soil modification?

We did not compare here type of MMT and chose one type – MMT-Na to comparison the intercalation degree (d-spacing) affected by type of organophilizer, however, our next work will be related to studies on an influence of type of MMT (with different cations) on physicochemical properties of Si-PSA.

  1. The image may have been improved.

We corrected the image, we hope it is correct now.

Finally, we hope that corrections made in the manuscript fulfill reviewer suggestions and allow editor to make positive decision about acceptation of our contribution for publishing in this journal.

With regards,

Adrian Krzysztof Antosik

Karolina Mozelewska

Magdalena Zdanowicz

Konrad Gziut

Piotr MiÄ…dlicki

Reviewer 2 Report

This paper details the organophilization of sodium montmorillonite using ammonium salts with various chain lengths and the influence of this to silicon pressure sensitive adhesives. The study contains the characterization of the organophilization using XRD, FT-IR, TGA and viscosity measurements as well as tests on adhesion, cohesion, tack and shrinkage.

The manuscript is well written and the findings presented are interesting, but the following minor issues should be addressed before publication.

1)    Several typos have to be corrected throughout the text (i.e line 19 przepared, line 20 udeful).

2)    The caption in figure 1 has to go after the figure.

3)    The figures 4-8 seems gray. They need correction.

Author Response

The authors would like to thank to the Reviewer and truly appreciate his comments, questions and corrections. Please find the detailed answers below.

This paper details the organophilization of sodium montmorillonite using ammonium salts with various chain lengths and the influence of this to silicon pressure sensitive adhesives. The study contains the characterization of the organophilization using XRD, FT-IR, TGA and viscosity measurements as well as tests on adhesion, cohesion, tack and shrinkage.

The manuscript is well written and the findings presented are interesting, but the following minor issues should be addressed before publication.

1)    Several typos have to be corrected throughout the text (i.e line 19 przepared, line 20 udeful).

We made a change. We hope it’s fine now.

2)    The caption in figure 1 has to go after the figure.

We made a change.

3)    The figures 4-8 seems gray. They need correction.

We made a change, we changed the colors of the markings. We hope the data will be more visible.

Finally, we hope that corrections made in the manuscript fulfill reviewer suggestions and allow editor to make positive decision about acceptation of our contribution for publishing in this journal.

With regards,

Adrian Krzysztof Antosik

Karolina Mozelewska

Magdalena Zdanowicz

Konrad Gziut

Piotr MiÄ…dlicki

Reviewer 3 Report

The article “Organophilized montmorillonites as fillers for silicone pressure-sensitive adhesives” presents insightful knowledge on the organophilize sodium montmorillonite (MMT) with selected ammonium salts and study the influence of OMMT on silicone pressure-sensitive adhesives (PSA).

My comments are:

·       The paper's originality is high, with a similarity index of 6% to the published work

·       The abstract is incomplete. It contains nothing. The abstract should include a brief background/introduction, objective, methods, quantitative results, discussion, and conclusion. It should be 200-300 words. Please improve it.

·       The state-of-the-art written in the introduction of the paper is improper and too brief. Please improve it.

·       The experimental procedure is justified and comprehensive according to this study.

·       The results are presented with proper TABLES and FIGURES.

·       Furthermore, compared to references, the results obtained are significant. The discussion section is clear and understandable and follows the results obtained.

·       The conclusion is not written properly.

·       The reference format is wrong. The authors should revise it according to the MATERIALS Journal.

·       The manuscript should go for English checking. Many grammatical errors were detected.

However, the article lacks major scientific soundness. Thus, the article can be considered after addressing the below comments in the revision (major revision).

Please check the specific comments below:

1.     Title: The title is relevant with the content. The authors can keep it as it is.

2.     Abstract:

a.     Many things are missing in the abstract. The abstract should include a brief introduction, methods, results, discussion, and conclusion. It should be in the range of 200-300 words. Revise this accordingly.

3.     Introduction:

a.     The introduction does not have a storyline. The research gap is missing in the introduction. The authors should revise this.

b.     Figure 1. The caption is misplaced. Revise this. Please provide the reference and the copyright for the figure.

c.     The authors should expand the introduction on using nanoclay as a filler/reinforcement agent in adhesives. The following articles could be used in the introduction:

·       Effects of nanoclay modification with transition metal ion on the performance of urea–formaldehyde resin adhesives. Polymer Bulletin. 78, 2375–2388 (2021). https://doi.org/10.1007/s00289-020-03214-3

·       Simultaneous Improvement of Formaldehyde Emission and Adhesion of Medium-Density Fiberboard Bonded with Low-Molar Ratio Urea-Formaldehyde Resins Modified with Nanoclay. Journal of the Korean Wood Science and Technology. 49 (5), 453-461 (2021). https://doi.org/10.5658/WOOD.2021.49.5.453

4.   Materials and Method:

a.     Page 3. Line 83. Revise to ‘BYK’.

b.     Section 2.1. The space and writing format are not wrong. Revise this.

c.     Remove 2.4. Methods. Just write directly ‘2.4. Montmorillonite modification’

d.     Page 3. Line 112. Revise to ‘dispersed’

e.     Please provide a ‘flow chart of the Modification of MMT clay’

·       For example:

f.      Page 4. Line 128. Please write the specific XRD method, such as the range of xxx°–xxx° and the step xxxx°/min. In addition, the specification of the XRD instrument is missing. Please write the type, manufacturer, and country. For example, XRD (D/Max-2500, Rigaku Miniflex II, Japan).

g.     Page 4. Line 133. Please write the country of the TGA instrument.

h.     Page 4. Line 139. FT-IR spectrophotometer (Perkin Elmer Spectrum 100, Waltham, MA, USA) is a good example.

i.      Page 4. Line 144. Please add references at the end of the pot life explanation.

j.      Page 4. Line 145. The viscometer used in the pot life measurement is missing. Did you check the viscosity manually or use a viscometer?

k.     Page 4. Line 157. Please write the specification of the PSTA coater.

l.      Page 5. Line 158. Please write ‘cross-linked’ consistently.

5.       Results and discussion:

a.     Table 1. Did the authors check the molecular weight? Or was it provided by the manufacturer? The method of molecular weight measurement is missing in the method. Please revise this accordingly.

b.     Figure 2. The FTIR spectra graphs are not suitable for publication. Revise this accordingly. The minimum resolution is 600 DPI for better clarity.

For example,

c.     The XRD of OMMT-PSA is missing. Please provide it.

6.     Conclusions

a.     Please expand the conclusion by adding a brief discussion of the results.

7.     References

Revise the references according to the MATERIALS Journal.

Author Response

The authors would like to thank to the Reviewer and truly appreciate his comments, questions and corrections. Please find the detailed answers below.

The article “Organophilized montmorillonites as fillers for silicone pressure-sensitive adhesives” presents insightful knowledge on the organophilize sodium montmorillonite (MMT) with selected ammonium salts and study the influence of OMMT on silicone pressure-sensitive adhesives (PSA).

My comments are:

  • The paper's originality is high, with a similarity index of 6% to the published work
  • The abstract is incomplete. It contains nothing. The abstract should include a brief background/introduction, objective, methods, quantitative results, discussion, and conclusion. It should be 200-300 words. Please improve it.

We have made corrections as suggested by the reviewer, we hope we did it right.

  • The state-of-the-art written in the introduction of the paper is improper and too brief. Please improve it.

We have made corrections as suggested by the reviewer, we hope we did it right.

  • The experimental procedure is justified and comprehensive according to this study.
  • The results are presented with proper TABLES and FIGURES.
  • Furthermore, compared to references, the results obtained are significant. The discussion section is clear and understandable and follows the results obtained.
  • The conclusion is not written properly.

We have made corrections as suggested by the reviewer, we hope we did it right.

  • The reference format is wrong. The authors should revise it according to the MATERIALS Journal.

We have made corrections according to the MATERIALS Journal, we hope we did it right.

  • The manuscript should go for English checking. Many grammatical errors were detected.

The manuscript has been sent to English checking, we hope that all errors and language glitches have been removed.

However, the article lacks major scientific soundness. Thus, the article can be considered after addressing the below comments in the revision (major revision).

Please check the specific comments below:

  1. Title: The title is relevant with the content. The authors can keep it as it is.

  1. Abstract:
  2. Many things are missing in the abstract. The abstract should include a brief introduction, methods, results, discussion, and conclusion. It should be in the range of 200-300 words. Revise this accordingly.

  We have made corrections as suggested by the reviewer, we hope we did it right.

  1. Introduction:
  2. The introduction does not have a storyline. The research gap is missing in the introduction. The authors should revise this.

  We have made corrections as suggested by the reviewer, we hope we did it right.

  1. Figure 1. The caption is misplaced. Revise this. Please provide the reference and the copyright for the figure.

We have made changes, drawing by myself, does not require copyright.

  1. The authors should expand the introduction on using nanoclay as a filler/reinforcement agent in adhesives. The following articles could be used in the introduction:
  • Effects of nanoclay modification with transition metal ion on the performance of urea–formaldehyde resin adhesives. Polymer Bulletin. 78, 2375–2388 (2021). https://doi.org/10.1007/s00289-020-03214-3

  • Simultaneous Improvement of Formaldehyde Emission and Adhesion of Medium-Density Fiberboard Bonded with Low-Molar Ratio Urea-Formaldehyde Resins Modified with Nanoclay. Journal of the Korean Wood Science and Technology. 49 (5), 453-461 (2021). https://doi.org/10.5658/WOOD.2021.49.5.453

   We have made corrections as suggested by the reviewer, we hope we did it right.

  1. Materials and Method:
  2. Page 3. Line 83. Revise to ‘BYK’.

We have made corrections as suggested by the reviewer, we hope we did it right.

  1. Section 2.1. The space and writing format are not wrong. Revise this.

We have made corrections as suggested by the reviewer, we hope we did it right.

  1. Remove 2.4. Methods. Just write directly ‘2.4. Montmorillonite modification’

We have made corrections as suggested by the reviewer, we hope we did it right.

  1. Page 3. Line 112. Revise to ‘dispersed’

We have made corrections as suggested by the reviewer, we hope we did it right.

  1. Please provide a ‘flow chart of the Modification of MMT clay’
  • For example:

 We have made provide as suggested by the reviewer, we hope we did it right.

  1. Page 4. Line 128. Please write the specific XRD method, such as the range of xxx°–xxx° and the step xxxx°/min. In addition, the specification of the XRD instrument is missing. Please write the type, manufacturer, and country. For example, XRD (D/Max-2500, Rigaku Miniflex II, Japan).

We have made corrections as suggested by the reviewer, we hope we did it right.

  1. Page 4. Line 133. Please write the country of the TGA instrument.

We have made corrections as suggested by the reviewer considering description h, we hope we did it right.

  1. Page 4. Line 139. FT-IR spectrophotometer (Perkin Elmer Spectrum 100, Waltham, MA, USA) is a good example.
  2. Page 4. Line 144. Please add references at the end of the pot life explanation.

We have made addition references.

  1. Page 4. Line 145. The viscometer used in the pot life measurement is missing. Did you check the viscosity manually or use a viscometer?

We have made addition specification of viscometer, we hope we did it right.

  1. Page 4. Line 157. Please write the specification of the PSTA coater.

We have made addition specification of coater, we hope we did it right.

  1. Page 5. Line 158. Please write ‘cross-linked’ consistently.

 We have made corrections as suggested by the reviewer, we hope we did it right.

  1. Results and discussion:
  2. Table 1. Did the authors check the molecular weight? Or was it provided by the manufacturer? The method of molecular weight measurement is missing in the method. Please revise this accordingly.

The molecular weight was provided by the manufacturer.

  1. Figure 2. The FTIR spectra graphs are not suitable for publication. Revise this accordingly. The minimum resolution is 600 DPI for better clarity.

For example,

Figure 1 and  2 have been corrected.

  1. The XRD of OMMT-PSA is missing. Please provide it.

There is too little filler content in the composite to see reflections on the XRD spectrum (reflections are hidden in the background).

  1. Conclusions
  2. Please expand the conclusion by adding a brief discussion of the results.

 We have made corrections as suggested by the reviewer, we hope we did it right.

  1. References

Revise the references according to the MATERIALS Journal.

 We have made corrections according to the MATERIALS Journal, we hope we did it right.

Finally, we hope that corrections made in the manuscript fulfill reviewer suggestions and allow editor to make positive decision about acceptation of our contribution for publishing in this journal.

With regards,

Adrian Krzysztof Antosik

Karolina Mozelewska

Magdalena Zdanowicz

Konrad Gziut

Piotr MiÄ…dlicki

Round 2

Reviewer 3 Report

The authors have improved the manuscript significantly.

However, the Figures 2 (XRD) and 3 (FTIR) are still presented with low-resolution figures. Please improve this.

Author Response

The authors would like to thank to the Reviewer and truly appreciate his comments, questions and corrections. Please find the detailed answers below.

The authors have improved the manuscript significantly.

However, the Figures 2 (XRD) and 3 (FTIR) are still presented with low-resolution figures. Please improve this.

We have made improves as suggested by the reviewer, we hope we did it right.

Finally, we hope that corrections made in the manuscript fulfill reviewer suggestions and allow editor to make positive decision about acceptation of our contribution for publishing in this journal.

With regards,

Adrian Krzysztof Antosik

Karolina Mozelewska

Magdalena Zdanowicz

Konrad Gziut

Piotr MiÄ…dlicki